# Unknown Subclinical Hypothyroidism and In-Hospital Outcomes and Short- and Long-Term All-Cause Mortality among ST Segment Elevation Myocardial Infarction Patients Undergoing Percutaneous Coronary Intervention

**DOI:** 10.3390/jcm9123829

**Published:** 2020-11-26

**Authors:** Elena Izkhakov, David Zahler, Keren-Lee Rozenfeld, Dor Ravid, Shmuel Banai, Yan Topilsky, Naftali Stern, Yona Greenman, Yacov Shacham

**Affiliations:** 1Institute of Endocrinology, Metabolism and Hypertension, Tel Aviv Sourasky Medical Center, Tel Aviv 6423906, Israel; naftalis@tlvmc.gov.il (N.S.); yonagr@tlvmc.gov.il (Y.G.); 2Sackler Faculty of Medicine, Tel Aviv University, Ramat Aviv 6997801, Israel; avidza@tlvmc.gov.il (D.Z.); kerenroz@tlvmc.gov.il (K.-L.R.); dorr@tlvmc.gov.il (D.R.); shmuelb@tlvmc.gov.il (S.B.); yant@tlvmc.gov.il (Y.T.); kobys@tlvmc.gov.il (Y.S.); 3Department of Cardiology, Tel Aviv Sourasky Medical Center, Tel Aviv 6423906, Israel

**Keywords:** subclinical hypothyroidism, ST elevation, myocardial infarction, percutaneous coronary intervention, thyroid function evaluation

## Abstract

Subclinical hypothyroidism (SCH) is defined as an elevated serum thyroid-stimulating hormone (TSH) level with a normal serum-free thyroxine (FT4) level. SCH has been associated with an increased risk of adverse cardiovascular outcomes. We investigated possible associations of unknown SCH with in-hospital outcomes and short- and long-term all-cause mortality in a large cohort of patients with ST segment elevation myocardial infarction (STEMI) who underwent primary percutaneous coronary intervention (PCI). This retrospective, single-center observational study evaluated the TSH and FT4 levels of 1593 STEMI patients with no known history of hypothyroidism or thyroid replacement treatment who were admitted to the coronary care unit and underwent PCI between 1/2008 and 8/2017. SCH was defined as TSH levels ≥ 5 mU/mL in the presence of normal FT4 levels. Unknown SCH was detected in 68/1593 (4.2%) STEMI patients. These patients had significantly worse in-hospital outcomes compared to patients without SCH, including higher rates of acute kidney injury (*p* = 0.003) and left ventricular ejection fraction ≤ 40% (*p* = 0.03). Moreover, 30-day mortality (*p* = 0.02) and long-term (mean 4.2 ± 2.3 years) mortality (*p* = 0.007) were also significantly higher in patients with SCH. The thyroid function of STEMI patients should be routinely tested before they undergo a planned PCI procedure.

## 1. Introduction

Subclinical hypothyroidism (SCH) is defined as a serum thyroid-stimulating hormone (TSH) level above the upper limits of normal, despite normal levels of serum-free thyroxine [1]. The prevalence of SCH in the general population is 3–8%, and the rate increases with age and is higher among women [2,3,4]. Individuals with overt hypothyroidism reportedly have an increased risk of functional cardiovascular abnormalities and coronary artery disease, as well as accelerated atherosclerosis [5,6]. Hypothyroidism is also associated with adverse outcomes among patients undergoing percutaneous coronary intervention (PCI) [7]. The presence of SCH is associated with lipid abnormalities and increased cardiovascular risk, particularly in older women [8,9,10]. The Thyroid Studies Collaboration found a significant increase in mortality from coronary heart disease among individuals with SCH, especially among those with levels of TSH ≥ 10.0 mIU/L [8]. Similar observations were reported for non-fatal and fatal myocardial infarction and heart failure events [8,10,11]. However, SCH among healthy individuals was not shown to be an independent risk factor for carotid atherosclerosis [12]. The present study was designed to investigate possible associations of unknown SCH with in-hospital outcomes and with short- and long-term all-cause mortality in a large historical cohort of Israeli patients with ST segment elevation myocardial infarction (STEMI) who underwent PCI.

## 2. Materials and Methods

### 2.1. Study Population and Design

This retrospective, single-center observational study was performed at the Tel-Aviv Sourasky Medical Center (TASMC), a tertiary referral hospital with a 24/7 primary PCI service. Those included in the study were all patients admitted between October 2007 and August 2017 with the diagnosis of acute STEMI, who were subsequently treated with primary PCI as described in detail elsewhere [13,14]. Those excluded were patients for whom data on TSH and free thyroxine fraction (FT4) levels were not available in their medical records, those who were treated with amiodarone, and those who had a documented history of hypothyroidism or hyperthyroidism.

There were no differences in the history of past or present major illnesses between the patients who were included in the study and those who were excluded.

The study protocol was approved by the TASMC ethics committee, which waived informed consent (Institutional Board Review Number: TLV-16-0224). All procedures performed in THIS study involving human participants were in accordance with the ethical standards of the institution and with the 1964 Helsinki declaration and its later modifications. 

### 2.2. Definitions of SCH, Overt Hypothyroidism and the Euthyroid State

Blood tests, including TSH and FT4 levels, were performed by immunoenzymatic methods (DPC, Los Angeles, CA, USA) upon admission to the cardiac intensive care unit following PCI. Normal values for thyroid function tests were 0.7–1.8 ng/dL for FT4 and 0.40–4.99 mIU/L for TSH. All the samples from all the patients were run in the same type of assay. SCH was defined as a TSH level of 5 mU/mL or more in the presence of a normal FT4 level [1]. Overt hypothyroidism was defined as a TSH level of 5 mU/mL or more in the presence of an FT4 level lower than 0.7 ng/dL, or treatment with thyroid replacement therapy, or antithyroid medications. A euthyroid state was defined by a TSH level within the normal range (0.5–4.99 mU/mL) in the presence of FT4 within the normal range (0.7–1.8 ng/dL).

### 2.3. Definition of STEMI Diagnosis

STEMI was diagnosed according to the relevant guidelines, including a typical history of chest pain, diagnostic electrocardiographic changes, and serial elevation of cardiac biomarkers [15]. According to the hospital protocol, a primary PCI was performed in patients with symptoms ≤ 12 h in duration, and in patients with symptoms lasting 12–24 h if the pain was present at the time of admission. Symptom duration was defined as the time from symptom onset (usually chest pain or discomfort) to emergency room/catheterization laboratory admission.

### 2.4. Definition of Time to Mortality Following PCI

Short-term all-cause mortality was defined as occurring up to 30 days after undergoing PCI. Long-term all-cause mortality data were available 4.2 ± 2.3 years after undergoing PCI.

### 2.5. Definitions of Chronic Kidney Disease and Acute Kidney Injury

The glomerular filtration rate (GFR) was estimated by means of the Chronic Kidney Disease Epidemiology Collaboration (CKD-EPI) equation [16]. Chronic kidney disease (CKD) was categorized as an admission GFR of <60 mL/min/1.73 m^2^. Acute kidney injury was determined by the Kidney Disease: Improving Global Outcomes criteria [17], and defined as either an increase in serum creatinine (sCr) ≥ 0.3 mg/dL within 48 h of admission or an increase in sCr ≥ 1.5 times that of baseline, which was known or presumed to have occurred within the preceding 7 days.

### 2.6. Data Collection and Clinical Follow-Up

The following data were retrieved from the hospital electronic medical records: baseline demographic characteristics, cardiovascular history, cardiovascular risk factors, treatment characteristics, and laboratory results (the latter including TSH, FT4, electrolytes, and kidney and liver function findings). Patients’ records were evaluated for 30-day mortality and complications occurring throughout hospitalization. These included cardiogenic shock and the need for intra-aortic balloon counterpulsation treatment, the need for mechanical ventilation, heart failure episodes occurring between hospital admission and discharge that were treated conservatively, clinically significant tachyarrhythmias, bradyarrhythmias requiring a pacemaker, major bleeds requiring blood transfusion, and episodes of acute kidney injury that occurred throughout hospitalization. Survival was assessed from computerized records of the population registry until study closure in August 2018, and survival data were available for all the patients included in the study.

### 2.7. Statistical Analysis

Continuous variables are given as means ± standard deviations, and they were compared with the independent sample t-test when normally distributed. Non-normally distributed continuous variables are given as medians and interquartile ranges, and they were compared with the Mann–Whitney U test. Categorical variables are presented as percentages, and *p* values were calculated with the chi-square test. Independent predictors of 30-day mortality were determined in a multivariate binary logistic regression model adjusted for all baseline variables found to be significant in the univariate analysis. Long-term survival rates were described by the Kaplan–Meier method. The Cox regression model was used to determine predictors of long-term mortality. A 2-tailed *p* value of <0.05 was considered significant for all analyses. All analyses were performed with the SPSS software (SPSS Inc., Chicago, IL, USA).

## 3. Results

### 3.1. Baseline Characteristics

A total of 2234 patients were admitted during the study period with the diagnosis of acute STEMI and subsequently treated with primary PCI. Of these 2234 patients, 589 were excluded from the analysis due to absent TSH level recordings, 45 due to thyroid hormone replacement therapy, and 7 with unknown clinical hypothyroidism based on low serum-free T4 levels. The remaining 1593 patients were included in the final analysis (median age 61 years, 82% males), and 68 (4.2%) of them demonstrated SCH upon admission to the cardiac intensive care unit. The baseline clinical characteristics of patients with SCH compared to those without SCH are presented in Table 1. Patients with SCH were more likely to be smokers and male sex.

### 3.2. In-Hospital Outcomes

Table 2 presents the in-hospital outcomes for patients with SCH compared to those without SCH. Patients with SCH were more likely to develop acute kidney injury and to have lower left ventricular ejection fraction (LVEF) levels. The rate at which new-onset atrial fibrillation developed during the hospitalization period was similar among both groups.

### 3.3. Short-Term Since PCI Mortality

Within 30 days of hospital admission, 6/68 (9%) patients with SCH died compared to 45/1525 (3%) without SCH (*p* = 0.02). A univariate logistic regression analysis revealed that the following factors were associated with short-term mortality: SCH, age over 60 years, and LVEF equal to or less than 40% (Table 3). SCH was independently associated with short-term (30-day) mortality in a multivariate binary logistic regression model adjusted for all baseline variables (odds ratio 3.2, 95% confidence interval [CI]: 1.2–8.6, *p* = 0.02). Age over 60 years, LVEF ≤ 40, and a family history of coronary artery disease were also significantly associated with 30-day mortality (Table 3).

### 3.4. Long-Term Mortality

In total, 16/68 (24%) patients with SCH and 202/1525 (13%) without SCH died (*p* < 0.001) over a median follow-up period of 4.2 years (interquartile range 2.2–6.5 years). Factors associated with long-term mortality included SCH, age over 60 years, hypertension, previous myocardial infarction, and LVEF equal to or less than 40% (Table 4). SCH was independently associated with long-term mortality following STEMI (hazard ratio 2.2, 95%CI: 1.2–3.8, *p* = 0.007, multivariable Cox regression model, Table 4, Figure 1).

## 4. Discussion

In this retrospective, single-center observational study, in-hospital outcomes and short- and long-term mortality were assessed among patients with STEMI and no known thyroid pathology who underwent primary PCI. The results demonstrated that previously unknown and untreated SCH was independently associated with poor in-hospital outcomes and higher short-term (30-day) and long-term (4.2 ± 2.3 years) mortality compared to euthyroid patients. These findings concur with a number of recent studies on adults with heart disease that showed poorer outcomes among those with co-existing SCH. Among Japanese patients with a mean age of 68 years who had heart failure, those with co-existing SCH had lower peak breath-by-breath oxygen consumption, higher mean pulmonary arterial pressure, and increased risk of all-cause mortality during a mean 3-year follow-up [18]. In a Korean study with 12 years of follow-up, increased cardiovascular events and mortality were reported among adults with both cardiovascular risk and SCH, and the risk was particularly high among patients under 65 years of age [19]. The results of another Korean study with a mean 8-year follow-up of patients after coronary artery bypass grafting showed a higher incidence of revascularization among those with SCH than those who were euthyroid [20]. Furthermore, a meta-analysis of 35 studies showed a modest risk of SCH with all-cause mortality in patients younger than 65 years of age, and that the risk was significantly increased in a subgroup analysis of those with a high cardiovascular risk [21]. Of particular relevance to the current study, an increased risk of repeat revascularization and cardiac death was reported among individuals with SCH compared to those who were euthyroid at 3 years after PCI [22].

Potential mechanisms for these findings in SCH patients include oxidative stress in mitochondria due to increased plasma inflammatory markers, insulin resistance, activation of thrombosis and hypercoagulability, endothelial dysfunction, delayed diastolic filling, impaired left ventricular systolic function, and increased vascular resistance [6,23,24,25,26,27,28,29]. In the current study, SCH was associated with an increased risk of acute kidney injury compared to euthyroid patients. These findings concur with various associations that have been documented between thyroid dysfunction and impaired kidney function, including in the setting of cardiovascular disease [30]. Notably, one recent hospital-based cross-sectional study reported higher creatinine levels and lower estimated glomerular filtration rates among patients with SCH than among euthyroid patients [31].

The patients with SCH in the current study were more likely to be males and smokers, and to have a low LVEF. Smoking and insulin resistance have been identified as factors that may modify the effects of SCH on lipid parameters [25]. Impairments in left ventricular diastolic function [24] as well as in right ventricular systolic and diastolic function are among the cardiac risk factors reportedly increased among individuals with SCH [32]. Notably, a meta-analysis of 14 studies of heart failure patients with reduced LVEF showed increased risk of all-cause mortality among those with SCH compared to those who were euthyroid [33]. In contrast to those findings and to the present ones, a recent study of heart failure patients found that LVEF did not differ between those with SCH and those who were euthyroid [18]. Moreover, the LVEF did not differ by thyroid fraction among individuals with acute coronary syndrome in another report [34].

The benefit of hormone replacement therapy for individuals with SCH is controversial. A number of controlled studies on hormone replacement therapy in this context showed improvements in endothelial function [23], in lipid profile, and in symptoms of tiredness [24]. However, application of the threshold of TSH to define and treat SCH remains inconclusive, and patient characteristics, such as the natural increase in TSH with age, regardless of the presence of thyroid disease, need to be considered as well [35,36,37]. A lower proportion of incident ischemic heart disease was observed among younger (aged 40–70 years) patients with SCH but not among older (aged over 70 years) patients with SCH treated with thyroid hormone replacement therapy [38]. A systematic review and meta-analysis of five placebo-controlled randomized controlled trials of individuals with SCH concluded that thyroid hormone therapy decreased TSH levels by 66% and low-density lipoprotein cholesterol levels by 14% [39], although these changes did not clearly indicate improved patient outcomes. A panel that included clinicians, patients, and methodologists from several countries worldwide recently issued guidelines against the administration of thyroid hormones to adults with SCH [40]. These recommendations were based on the evidence of a systematic review that identified 21 trials with over 2000 participants that did not show any beneficial effects of thyroid hormones on quality of life, cardiovascular events, or mortality [41].

### Limitations

This study has several limitations that bear mentioning, starting with its single-center, non-randomized, and observational design. The number of patients with SCH is relatively small, and the number of events in that group is small as well. Stringent risk adjustment was employed in order to mitigate these effects, but the possibility of residual confounding by other non-measured factors cannot be excluded. In addition, this study only included patients who were undergoing primary PCI, and the results cannot be generalized to all STEMI patients. Information on further diagnostics (and eventual treatment) of hypothyroidism during hospital stay and follow-up, as well as on control measures of TSH and FT4, is not present in the current database. The period of time covered is 10 years, which may be relevant since the important diagnostics and treatment changes that occurred during that decade might have affected outcomes. Data on past medications used and those recommended upon discharge were not available in the current database, thus their effect could not have been assessed.

## 5. Conclusions

Unknown SCH is not uncommon among STEMI patients treated with PCI, and it may serve as a prognostic marker for poor in-hospital outcomes and elevated short- and long-term mortality. These findings are important because the prevalence of SCH increases with age, as does the presentation of STEMI. Thyroid function tests are not routinely performed during hospitalization, and the current findings indicate that the information obtained from them on in-hospital outcomes and short- and long-term mortality is highly relevant to the planned performance of PCI. Therefore, routine testing of thyroid function before performing PCI should be considered, and prospective studies are warranted to clarify the optimal management for STEMI patients with SCH who are planned for PCI.

## Figures and Tables

**Figure 1 jcm-09-03829-f001:**
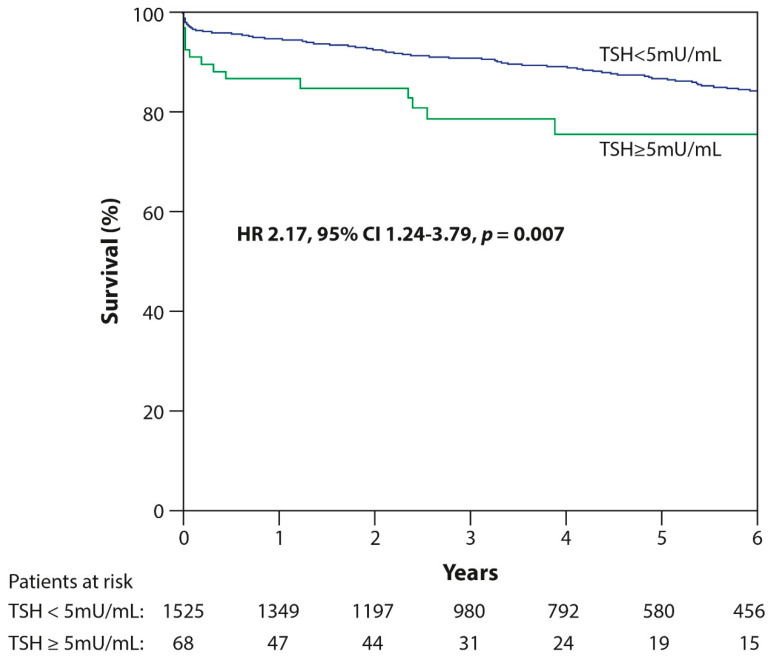
Kaplan–Meier Curve for Long-Term Survival of Patients with Subclinical Hypothyroidism vs. Euthyroid Patients. Green: patients with subclinical hypothyroidism (TSH > 5 mU/mL in the presence of a normal FT4 level); Blue: euthyroid patients (normal thyroid function, TSH < 5 mU/mL). TSH, thyroid-stimulating hormone; HR, hazard ratio; CI, confidence interval.

**Table 1 jcm-09-03829-t001:** Baseline criteria of 1593 STEMI patients according to the presence of subclinical hypothyroidism.

Variable	Euthyroid Patients (*n* = 1525)	Patients with SCH (*n* = 68)	*p* Value
Age, years, mean ± SD	61 ± 7	62 ± 8	0.41
Age > 60 years, *n* (%)	772 (51)	36 (53)	0.71
Male sex, *n* (%)	1258 (83)	40 (59)	**<0.001**
Diabetes mellitus, *n* (%)	367 (24)	20 (29)	0.31
Hyperlipidemia, *n* (%)	745 (49)	40 (59)	0.11
Family history of CAD, *n* (%)	325 (21)	16 (24)	0.67
Smoking, *n* (%)	748 (49)	44 (65)	**0.012**
Hypertension, *n* (%)	668 (44)	33 (49)	0.45
Chronic kidney disease (eGFR ≤ 60), *n* (%)	378(25)	22(32)	0.16
Multivessel coronary disease, *n* (%)	862(57)	39(59)	0.73
Past myocardial infarction, *n* (%)	193 (13)	12 (18)	0.23
Time to ER, minutes, median (IQR 25–75)	120 (60–360)	120 (60–280)	0.81
Admission CRP, mg/dl, median (IQR 25–75)	4.4 (1.5–10.9)	6.6 (1.7–13.3)	0.34
Duration of hospitalization, days, mean ± SD	6 ± 5	7 ± 11	0.27

STEMI, ST segment elevation myocardial infarction; SCH, subclinical hypothyroidism (as determined by TSH > 5 mu/L in the presence of a normal FT4 level); SD, standard deviation; CAD, coronary artery disease; eGFR, estimated glomerular filtration rate; ER, emergency room; IQR, interquartile range; CRP, C-reactive protein. **Bold** indicates significant.

**Table 2 jcm-09-03829-t002:** In-hospital outcomes of 1593 STEMI patients according to the presence of subclinical hypothyroidism.

Variable	Euthyroid Patients (*n* = 1525)	Patients with SCH (*n* = 68)	*p* Value
Heart failure, *n*	153 (10%)	7 (10%)	0.95
Acute kidney injury, *n*	146 (10%)	14 (20%)	**0.003**
LVEF, mean ± SD	47 ± 8	44 ± 9	**0.014**
LVEF ≤ 40%, *n*	452 (30%)	29 (43%)	**0.03**
Bleeding, *n*	62 (4%)	5 (7%)	0.20
Ventricular tachycardia/fibrillation, *n*	116 (8%)	8 (12%)	0.21
Bradycardia, *n* (%)	66 (4%)	3 (4%)	0.99
Intra-aortic balloon counterpulsation, *n*	64 (4%)	5 (7%)	0.21
New-onset atrial fibrillation, *n*	79 (5%)	0 (0)	**0.05**
In-hospital CABG, *n*	26 (2%)	2 (3%)	0.34
Mechanical ventilation, *n*	74 (5%)	6 (9%)	0.15
30-day mortality, *n*	45 (3%)	6 (9%)	**0.02**

LVEF, left ventricular ejection fraction; SCH, subclinical hypothyroidism (as determined by TSH > 5 mu/L in the presence of a normal FT4 level); SD, standard deviation; CABG, coronary artery bypass grafting. **Bold** indicates significant.

**Table 3 jcm-09-03829-t003:** Univariate and multivariate binary logistic regression model for 30-day mortality according to the presence of subclinical hypothyroidism.

Variable	Univariate		Multivariate	
	OR (95%CI)	*p* Value	OR (95%CI)	*p* Value
TSH ≥ 5 mU/mL	3.2 (1.3–7.7)	**0.01**	3.2 (1.2–8.6)	**0.02**
Female sex	3.2 (1.8–5.8)	**<0.001**	1.8 (0.9–3.4)	0.09
Age > 60 years	5.4 (2.5–11.6)	**<0.001**	2.9 (1.2–6.9)	**0.02**
Hypertension	2.2 (1.2–3.9)	**0.01**	1.4 (0.7–2.4)	0.29
LVEF ≤ 40%	9.7 (4.6–20.4)	**<0.001**	4.3 (1.9–9.8)	**0.001**
Family history of CAD	0.2 (0.1–0.7)	**0.01**	0.3 (0.7–1.34)	0.12
Smoking	0.4 (0.2–0.7)	**0.002**	0.6 (0.3–1.3)	0.23
Diabetes mellitus	1.4 (0.8–2.6)	0.23	ND	ND
Hyperlipidemia	0.8 (0.4–1.4)	0.37	ND	ND

OR, odds ratio; CI, confidence interval; TSH, thyroid-stimulating hormone; LVEF, left ventricular ejection fraction; CAD, coronary artery disease; ND, not done due to lack of significance in the univariate analysis. **Bold** indicates significant.

**Table 4 jcm-09-03829-t004:** Univariate and multivariate Cox regression model predicting long-term mortality according to the presence of subclinical hypothyroidism.

Variable	Univariate		Multivariate	
	HR (95%CI)	*p* Value	HR (95%CI)	*p* Value
TSH ≥ 5 mU/mL	2.2 (1.3–3.6)	**0.003**	2.2 (1.2–3.8)	**0.007**
Female sex	2.1 (1.5–2.8)	**<0.001**	1.2 (0.8–1.5)	0.36
Age > 60 years	7.0 (4.8–10.3)	**<0.001**	4.5 (2.9–6.8)	**0.001**
Hypertension	2.8 (2.1–3.7)	**<0.001**	1.8 (1.3–2.4)	**0.001**
Diabetes mellitus	1.7 (1.3–2.2)	**<0.001**	1.2 (0.8–1.6)	0.35
LVEF ≤ 40%	2.2 (1.7–2.9)	**<0.001**	1.4 (1.1–1.9)	**0.02**
Past myocardial infarction	2.0 (1.4–2.8)	**<0.001**	1.6 (1.1–2.3)	**0.01**
Family history of CAD	0.2 (0.1–0.3)	**<0.001**	0.3 (0.1–0.6)	**0.001**
Smoking	0.5 (0.4–0.7)	**<0.001**	0.9 (0.7–1.3)	0.86
Hyperlipidemia	1.1 (0.8–1.4)	0.48	ND	ND

HR, hazard ratio; CI, confidence interval; TSH, thyroid-stimulating hormone; LVEF, left ventricular ejection fraction; CAD, coronary artery disease; ND, not done due to lack of significance in the univariate analysis. **Bold** indicates significant.

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
