# Peer review of "Unknown Subclinical Hypothyroidism and In-Hospital Outcomes and Short- and Long-Term All-Cause Mortality among ST Segment Elevation Myocardial Infarction Patients Undergoing Percutaneous Coronary Intervention"

_jcm, 2020, doi:10.3390/jcm9123829_

Round 1
Reviewer 1 Report
In this retrospective single-center observational study, the authors found that subclinical hypothyroidism in STEMI patients who underwent PCI was closely associated with in-hospital, 30-day and long-term all-cause mortality. Although well-written in general, this reviewer has some comments as follows.
1. The authors need to clarify whether there is a systemic difference between the study population and the excluded population due to lack of TSH level.
2. TSH was tested in about 74% of patients with STEMI at the time of admission. Could the authors explain why? In general practice, it is not common to test TSH in patients with acute coronary syndrome or acute MI.
3. The number of patients with SCH group is relatively too small, and the number of events in that group also is too small. It may be considered insufficient to evaluate the statistical significance through multivariate adjustment. The authors should discuss this issue. Also, the graphical presentation through the Kaplan-Meier survival curve for the outcomes by each group may be helpful in interpretating the results of this study.
4. Is there any information on other cardiovascular events (e.g., MI, or repeat revascularization) during the short- and long-term follow-ups? All-cause mortality is presented as the primary endpoint, but it is not known whether the death was due to cardiovascular cause, which also limits the interpretation of results.
5. Although the authors described in the limitation section, I should mention that many important confounding factors that can affect mortality in STEMI were omitted, and I think variables that should be judged as baseline features were incorrectly classified as in-hospital outcomes. When did the heart failure or acute kidney injury occur? Was it present at the time of admission, or occurred after the admission? Is there any information on heart failure or renal failure that was already present at the time of baseline admission? Authors should present these additional information in Table 1, and also should consider to include these variables in the multivariate model. In addition, information on medications previously being taken, information related to the severity of coronary artery disease (left main or multivessel CAD, number of implanted stents or number of treated lesions, etc.) were also omitted. I think the authors should clarify this.
6. The authors should be careful to derive the conclusion from the study results. Although this study showed that SCH was a significant predictor of post-PCI mortality in STEMI patients, this cannot support the argument for recommending routine thyroid function tests in STEMI patients. In order to make this logical, it is necessary to check whether discriminant function is improved or reclassification is improved when SCH is added to the model including classical variables known to predict mortality.
Author Response
Reviewer 1
- The authors need to clarify whether there is a systemic difference between the study population and the excluded population due to lack of TSH level.
Response:
This was now clarified in the Methods section: "There were no differences in the history of past or present major illnesses between the patients who were included in the study and those were excluded." (Methods, page 4).
- TSH was tested in about 74% of patients with STEMI at the time of admission. Could the authors explain why? In general practice, it is not common to test TSH in patients with acute coronary syndrome or acute MI.
Response:
TSH was initially (2007-2010) tested at admission upon decision of the medical personnel. From 2011 onwards, testing for TSH was routinely added to blood testing of all patients admitted to the CICU.
- The number of patients with SCH group is relatively too small, and the number of events in that group also is too small. It may be considered insufficient to evaluate the statistical significance through multivariate adjustment. The authors should discuss this issue. Also, the graphical presentation through the Kaplan-Meier survival curve for the outcomes by each group may be helpful in interpretating the results of this study.
Response:
As advised, the following was added to the Limitation section: "The number of patients with SCH is relatively small, and the number of events in that group is small as well. Stringent risk adjustment was employed in order to mitigate these effects, but the possibility of residual confounding by other non-measured factors cannot be excluded." (Limitations, page 10).
The suggested graphical presentation is now included in the paper (Figure 1).
- Is there any information on other cardiovascular events (e.g., MI, or repeat revascularization) during the short- and long-term follow-ups? All-cause mortality is presented as the primary endpoint, but it is not known whether the death was due to cardiovascular cause, which also limits the interpretation of results.
Response:
Our analysis is based upon retrospective data, collected over many years and it now includes over 3000 patients with STEMI admitted to our department. We agree entirely that the information requested by the reviewer regarding other short- and long-term outcomes (recurrent MI, revascularization) is important but, unfortunately, it not available in the current database.
- Although the authors described in the limitation section, I should mention that many important confounding factors that can affect mortality in STEMI were omitted, and I think variables that should be judged as baseline features were incorrectly classified as in-hospital outcomes. When did the heart failure or acute kidney injury occur? Was it present at the time of admission, or occurred after the admission? Is there any information on heart failure or renal failure that was already present at the time of baseline admission? Authors should present this additional information in Table 1, and also should consider to include these variables in the multivariate model. In addition, information on medications previously being taken, information related to the severity of coronary artery disease (left main or multivessel CAD, number of implanted stents or number of treated lesions, etc.) were also omitted. I think the authors should clarify this.
Response:
Heart failure episodes occurred between hospital admission and discharge.
This was now clarified in the Methods section (page 5):
"Patients' records were evaluated for 30-day mortality and complications occurring throughout hospitalization. These included cardiogenic shock and the need for intra-aortic balloon counterpulsation treatment, the need for mechanical ventilation, heart failure episodes occurring between hospital admission and discharge that were treated conservatively,"
Renal failure upon admission was referred to as CKD if the eGFR was <60 ml/ min/1.73 m². This was now clarified in the Methods section (page 5):
"The glomerular filtration rate (GFR) was estimated by means of the Chronic Kidney Disease Epidemiology Collaboration (CKD-EPI) equation.16 Chronic kidney disease (CKD) was categorized as an admission GFR of <60 ml/ min/1.73 m²."
Acute kidney injury occurred during hospitalization. This was now clarified in the Methods section (page 5):
"The AKI was determined by the KDIGO criteria,17 and defined as either an increase in sCr ≥0.3 mg/dl within 48 hours of admission or an increase in sCr ≥1.5 times that of baseline, which was known or presumed to have occurred within the preceding 7 days."
And (page 5-6) "… episodes of acute kidney injury that occurred throughout hospitalization."
Information on medications is not present in the current database (added to limitations, page 11), however, information on CAD severity was obtainable and it has now been added to Table 1.
- The authors should be careful to derive the conclusion from the study results. Although this study showed that SCH was a significant predictor of post-PCI mortality in STEMI patients, this cannot support the argument for recommending routine thyroid function tests in STEMI patients. In order to make this logical, it is necessary to check whether discriminant function is improved or reclassification is improved when SCH is added to the model including classical variables known to predict mortality.
Response:
As recommended, we have "softened" the conclusion to read as follows:
"Routine testing of thyroid function before performing PCI should, therefore, be considered, and prospective studies are warranted to clarify the optimal management for STEMI patients with SCH who are planned for PCI."(Conclusions, page 11).
Reviewer 2 Report
Specific comments:
- This is single center retrospective study what is important limitation.
- The period of time covered is 10 years. This may be important since important diagnostics and treatment changes occurred during these years.
- About 25% of patients were excluded due to lack of TSH results. What was the routine approach in STEMI patients concerning decision of TSH/FT4 lab assessment? Was it based on anamnesis? Routine?
- Median follow-up period was 4.2 years but with interquartile range 2.2-6.5 years. It would be interesting to see Kaplan-Meier curves.
- Please add, if possible, data concerning further diagnostics (and eventual treatment) of hypothyr during hospital stay and follow up if done in analyzed patients.
- Were any control measures of TSH, FT4 done in those patients?
- What is practical value of presented results?
Author Response
Reviewer 2
- This is single center retrospective study what is important limitation.
Response:
The following was added to the limitations section:
"This study has several limitations that bear mention, starting with its single-center, non-randomized, and observational design" (Limitations, page 10).
- The period of time covered is 10 years. This may be important since important diagnostics and treatment changes occurred during these years.
Response:
Our thanks for pointing out this issue. We added the following to the limitations:
"The period of time covered is 10 years, and that may be relevant since the important diagnostics and treatment changes that occurred during that decade might have affected outcomes." (Limitations, pages 10-11).
- About 25% of patients were excluded due to lack of TSH results. What was the routine approach in STEMI patients concerning decision of TSH/FT4 lab assessment? Was it based on anamnesis? Routine?
Response:
TSH was initially (2007-2010) tested at admission upon decision of the medical personnel. From 2011 onwards, testing for TSH was routinely added to blood testing of all patients admitted to the CICU.
- Median follow-up period was 4.2 years but with interquartile range 2.2-6.5 years. It would be interesting to see Kaplan-Meier curves.
Response:
A Kaplan-Meier curve was now added, with thanks.
- Please add, if possible, data concerning further diagnostics (and eventual treatment) of hypothyr during hospital stay and follow up if done in analyzed patients. Were any control measures of TSH, FT4 done in those patients?
Response:
Unfortunately, information on further diagnostics (and eventual treatment) of hypothyroidism during hospital stay, follow-up and control measures of TSH, FT4 is not present in the current database. This was now added to the Limitations section (page 10):
" Information on further diagnostics (and eventual treatment) of hypothyroidism during hospital stay and follow-up, as well as on control measures of TSH, FT4 is not present in the current database."
- What is practical value of presented results?
Response:
We suggested the practical value of our findings in the Conclusion section (page 11) "Unknown SCH is not uncommon among STEMI patients treated with PCI, and it may serve as a prognostic marker for poor in-hospital outcomes and elevated short- and long-term mortality. These findings are important because the prevalence of SCH increases with age, as does the presentation of STEMI. Thyroid function tests are not routinely performed during hospitalization, and the current findings indicate that the information obtained from them on in-hospital outcomes and short- and long-term mortality are highly relevant to the planned performance of PCI."
Round 2
Reviewer 1 Report
I don't have any additional comment, and I think the authors provided appropriate revision to sufficiently improve the quality of the overall manuscript.